

# Assessment of the bacterial community structure in shallow and deep sediments of the Perdido Fold Belt region in the Gulf of Mexico

Ma. Fernanda Sánchez-Soto Jiménez[1], Daniel Cerqueda-García[2], Jorge L. Montero-Muñoz[1], Ma. Leopoldina Aguirre-Macedo[1] and José Q. García-Maldonado[3]

[1] Centro de Investigación y de Estudios Avanzados del Instituto Politécnico Nacional, Unidad Mérida, Departamento de Recursos del Mar, Mérida, Yucatán, México
[2] Consorcio de Investigación del Golfo de México (CIGOM). Centro de Investigación y de Estudios Avanzados del Instituto Politécnico Nacional, Unidad Mérida. Departamento de Recursos del Mar, Mérida, Yucatán, México
[3] CONACYT - Centro de Investigación y de Estudios Avanzados del Instituto Politécnico Nacional, Unidad Mérida. Departamento de Recursos del Mar, Mérida, Yucatán, México

Corresponding authors
Ma. Leopoldina Aguirre-Macedo, leopoldina.aguirre@cinvestav.mx, leopoldina2305@gmail.com
José Q. García-Maldonado, jose.garcia@cinvestav.mx

## ABSTRACT

The Mexican region of the Perdido Fold Belt (PFB), in northwestern Gulf of Mexico (GoM), is a geological province with important oil reservoirs that will be subjected to forthcoming oil exploration and extraction activities. To date, little is known about the native microbial communities of this region, and how these change relative to water depth. In this study we assessed the bacterial community structure of surficial sediments by high-throughput sequencing of the 16S rRNA gene at 11 sites in the PFB, along a water column depth gradient from 20 to 3,700 m, including five shallow (20–600 m) and six deep (2,800–3,700 m) samples. The results indicated that OTUs richness and diversity were higher for shallow sites (OTUs $= 2{,}888.2 \pm 567.88$; $H' = 9.6 \pm 0.85$) than for deep sites (OTUs $= 1{,}884.7 \pm 464.2$; $H' = 7.74 \pm 1.02$). Nonmetric multidimensional scaling (NMDS) ordination revealed that shallow microbial communities grouped separately from deep samples. Additionally, the shallow sites plotted further from each other on the NMDS whereas samples from the deeper sites (abyssal plains) plotted much more closely to each other. These differences were related to depth, redox potential, sulfur concentration, and grain size (lime and clay), based on the environmental variables fitted with the axis of the NMDS ordination. In addition, differential abundance analysis identified 147 OTUs with significant fold changes among the zones (107 from shallow and 40 from deep sites), which constituted 10 to 40% of the total relative abundances of the microbial communities. The most abundant OTUs with significant fold changes in shallow samples corresponded to *Kordiimonadales, Rhodospirillales*, *Desulfobacterales* (*Desulfococcus*), *Syntrophobacterales* and *Nitrospirales* (*GOUTA 19, BD2-6, LCP-6*), whilst *Chromatiales, Oceanospirillales* (*Amphritea, Alcanivorax*), *Methylococcales*, *Flavobacteriales, Alteromonadales* (*Shewanella, ZD0117*) and *Rhodobacterales* were the better represented taxa in deep samples. Several of the OTUs detected in both deep and shallow sites have been previously related to hydrocarbons consumption. Thus, this metabolism seems to be well represented in the studied sites, and it could abate future hydrocarbon contamination in this ecosystem. The results presented herein,

along with biological and physicochemical data, constitute an available reference for further monitoring of the bacterial communities in this economically important region in the GoM.

## INTRODUCTION

Microorganisms are well recognized as key drivers of biogeochemical cycles in marine environments (*Webster et al., 2003*; *Santos et al., 2011*). Measuring the changes in microbial communities is of particular interest to understanding how environmental factors modulate their structure and how that, in turn, is related to the function and stability of the ecosystem (*Huber et al., 2007*; *Jones et al., 2012*; *Fuhrman, Cram & Needham, 2015*). In marine sediments, the type of substrate, energy and carbon sources, and the variables that are correlated with water depth (e.g., temperature, oxygen concentration, light penetration and hydrostatic pressure), generally describe the most important aspects of the variability amongst habitats (*Miller & Wheeler, 2012*).

In the Gulf of Mexico (GoM), changes in bacterial community structure have been mainly related with depth in the water column, and likely result from differences in temperature, dissolved oxygen and suspended particles (*King et al., 2013*) which occur across these depth differences. In marine sediments from the GoM, bacterial community composition has been determined at different depths below the seafloor (from sediment cores). It has been proposed that the bacterial community composition of these sediments likely results from the interaction between the water column and a benthic microbial population limited to the upper layer of the sediments (*Reese et al., 2013*). In contrast, the microbial diversity present in different sediment depths from seep systems in the GoM have been directly related to the composition and magnitude of hydrocarbon seepage (e.g., natural oils, methane, and non-methane hydrocarbons) (*Orcutt et al., 2010*), as well as to the presence of overlying microbial mats (*Mills et al., 2004*).

Knowledge about microbial communities in the GoM increased notably with the Deepwater Horizon (DWH) massive oil spill in 2010, which occurred in the north zone of the GoM across an enormous area with different environmental conditions (*Kostka et al., 2011*; *Mason et al., 2012*; *Liu & Liu, 2013*; *Joye, Teske & Kostka, 2014*). During the spill, it was observed that different bacterial phyla in the deep-water plume (e.g., *Oceanospirillales*, *Cycloclasticus* and *Colwellia*) rapidly responded and were enriched within hours - weeks following the DWH well blow out (*Redmond & Valentine, 2011*; *Mason et al., 2012*). In natural hydrocarbon seep sites sampled at the same time, several rare taxa increased rapidly in abundance after the spill, emphasizing the importance of specialized sub-populations and potential ecotypes during massive deep-sea oil discharges (*Kleindienst et al., 2015*). The oil transported to the shoreline after the discharge also had a profound impact on the abundance and community composition of indigenous bacteria in beach sands, where
members of the *Gammaproteobacteria* and *Alphaproteobacteria* participated as key players in oil degradation (*Kostka et al., 2011*).

The precise volume of oil spilled and the trajectory of the oil slicks from the DHW are still controversial (*Salcedo et al., 2017*). Based on the surface circulation models of the waters of the GoM, it was inferred that the Mexican region of the Perdido Fold Belt (PFB), was also susceptible to this environmental disturbance (*Soto & Vázquez-Botello, 2013*). Knowledge of environmental and biotic data of this region is, however, scarce despite being a geologic province with oil reservoirs. The present study assessed changes in the bacterial community structure of surficial sediments with respect to a depth gradient and specific environmental variables in the PFB region, where oil exploration and extraction activities are predicted to impact environmental conditions, and consequently the structure of bacterial communities. Additionally, bacterial phylotypes putatively involved in hydrocarbon degradation were highlighted in this work.

## MATERIAL AND METHODS

### Sample collection and physicochemical variables from marine sediments

The samples used in this study were collected in the Northwestern Gulf of Mexico in April 2014. Sediment samples from a depth gradient (20–3,700 m), were collected perpendicular to the coastline at 11 sites on the parallels 25 and 25.15°N (Fig. 1, Table 1). Samples were collected with a Hessler-Sandia MK-II boxcore ($40 \times 40$ cm) from which three different surficial (0–5 cm) subsamples were taken: (1) sterile 100 mL plastic containers immediately frozen at $-20$ °C on board for further molecular analysis; (2) a 2 inch core extracted to determine total sulfur concentration (TS) and redox potential on board, with a sulfide ion selective electrode and potentiometer (*Bricker, 1982*; *Brassard, 1997*); and (3) approximately 400 g of sediment stored in high density polyethylene bags at 4 °C until the determination of total organic matter (TOM), total organic carbon (TOC), and grain size in the laboratory. TOM was determined by the wet oxidation technique with an excess of dichromate and back titration with iron (II) (*Buchanan, 1984*), TOC content was quantified using the oxidation with potassium dichromate in acid medium, and titration of excess oxidant with ferrous sulfate and diphenylamine as an indicator (*Buchanan, 1984*). The amounts of sand, lime and clay were estimated using a hydrometer (*Buchanan, 1984*). These physicochemical variables were compared among shallow and deep sites using a one-way analysis of variance (ANOVA $p < 0.05$ and $F$-values) and were correlated with the depth variable.

### DNA extraction

Sediment samples were stored during five months at $-20$ °C. Then, in October 2014 sediments were thawed at 4 °C, homogenized mechanically under sterile conditions, and centrifuged 1 min/10,000 $\times$g in order to separate the sediment from remaining water, which was discarded. Total DNA was extracted from 0.25 g of sediment using the PowerSoil®DNA Isolation Kit (Mo Bio Laboratories, Carlsbad, CA, USA) following the manufacturer's protocol. The quality of the DNA extractions was verified by agarose gel

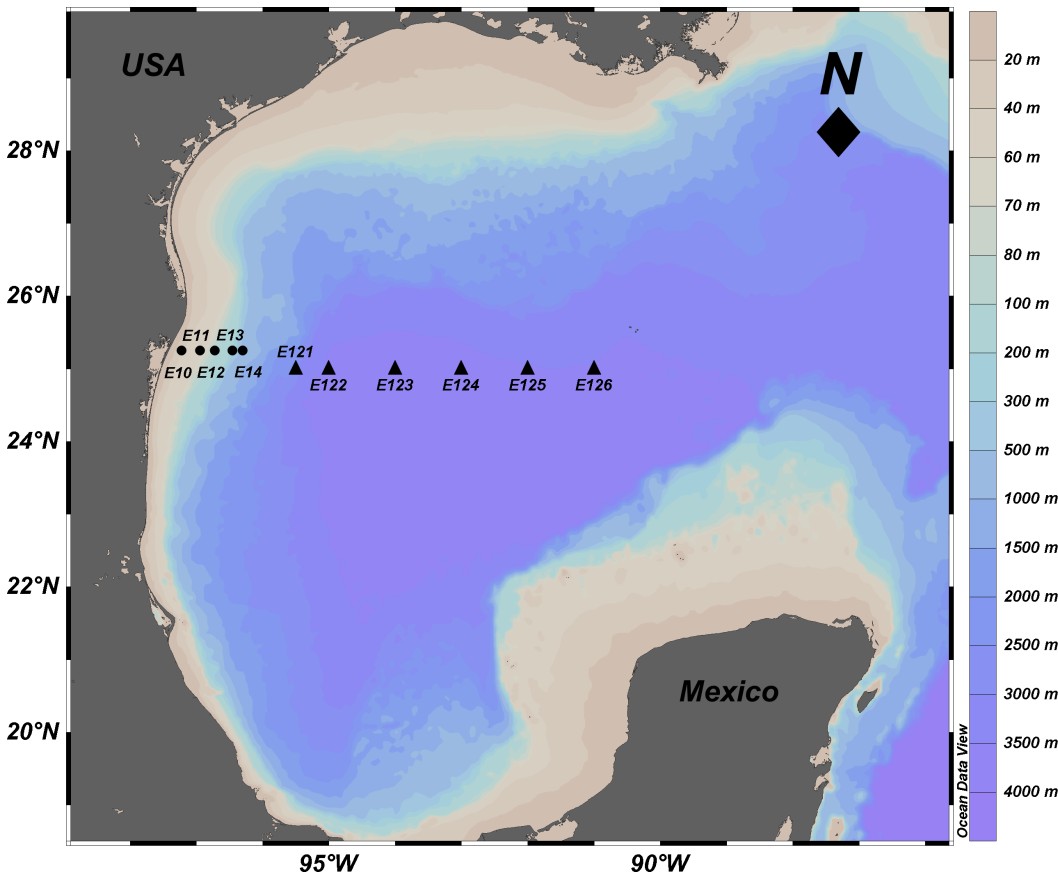

**Figure 1** **Location of sampling sites in the Mexican Region of the Northwestern Gulf of Mexico.**
Circles: shallow sites in the continental platform and slope from 20 to 600 m depth. Triangles: deep sites in
the abyssal plain from 2,800 to 3,700 m depth. The names of the stations are indicated. Source credit: Abril
Gamboa-Muñoz.

electrophoresis and the concentration was determined through UV absorption analysis
with a NanoDrop 2000 Spectrometer (ThermoFisher Scientific Inc., Wilmington, DE,
USA). Extracted DNA was stored at $-20\,°C$ for further 16S rRNA Illumina sequencing
and DGGE analysis (see a detailed description in Supplemental Information 1).

## 16S rRNA gene sequencing

Amplicons from environmental DNA were prepared for sequencing the 16S rRNA
V3 and V4 variable regions by using a two PCR steps approach following the "16S
Metagenomic Sequencing Library Preparation" protocol (Illumina). Briefly, the
first PCR step amplified the template out of the DNA samples using the forward primer
5′-CCTACGGGNGGCWGCAG-3′ and the reverse primer 5′-GACTACHVGGGTATCTAA
TCC-3′ with Illumina overhang adapters attached, to obtain ∼550 bp fragments
(*Klindworth et al., 2013*). The PCR program was performed in a thermal cycler (Applied
Biosystems Veriti ABI Inc., Foster City, CA, USA) with an initial denaturation at 95 °C–
3 min, 25 cycles of 95 °C–30 s, 55 °C–30 s, 72 °C–30 s and a final extension at 72 °C–5 min.

**Table 1  Physicochemical variables from shallow and deep sediment samples.**

| Site | Sample | Geographical location | | Physicochemical variables | | | | | | | |
| | | Longitude | Latitude | Depth (m) | Redox ↑ (mV) | TS ↑ (µM) | TOM (%) | TOC (µM) | Sand (%) | Lime ↑ (%) | Clay ↓ (%) |
|---|---|---|---|---|---|---|---|---|---|---|---|
| Shallow | E10 | 97°13′48″W | 25°15′N | 20 | −131.0 | 0.05 | 0.80 | 0.45 | 34.5 | 51.2 | 14.4 |
| | E11 | 96°56′24″W | 25°15′N | 75 | −174.4 | 0.05 | 1.10 | 0.61 | 26.5 | 57.3 | 16.3 |
| | E12 | 96°43′12″W | 25°15′N | 100 | −204.0 | 0.05 | 1.02 | 0.56 | 20.5 | 51.2 | 28.4 |
| | E13 | 96°27′36″W | 25°15′N | 200 | −131.7 | 0.05 | 0.83 | 0.46 | 22.5 | 51.2 | 26.4 |
| | E14 | 96°18′36″W | 25°15′N | 600 | −192.9 | 0.05 | 1.02 | 0.56 | 23.7 | 68.2 | 8.1 |
| Deep | E121 | 95°30′W | 25°00′N | 2,800 | 230 | 0.10 | 1.22 | 0.68 | 33.75 | 60.3 | 6.0 |
| | E122 | 95°00′W | 25°00′N | 3,600 | 233 | 0.10 | 0.59 | 0.33 | 27.75 | 66.3 | 6.0 |
| | E123 | 94°00′W | 25°00′N | 3,700 | 225 | 0.11 | 0.44 | 0.25 | 25.7 | 66.3 | 8.0 |
| | E124 | 93°00′W | 25°00′N | 3,700 | 215 | 0.11 | 0.85 | 0.47 | 29.8 | 64.3 | 5.9 |
| | E125 | 92°00′W | 25°00′N | 3,500 | 236 | 0.12 | 0.81 | 0.45 | 23.9 | 70.3 | 5.9 |
| | E126 | 91°00′W | 25°00′N | 3,700 | 215 | 0.12 | 0.92 | 0.51 | 25.8 | 68.2 | 6.0 |

**Notes.**

*TS*, total sulfur; *TOM*, total organic matter and *TOC*, total organic carbon. Physicochemical variables measured with no replicates. Pearson coefficient showed significant correlation at a *p*-value <0.01 among TS ($r^2 = 0.98$), redox potential ($r^2 = 0.98$), and the percentages of lime ($r^2 = 0.78$) and clay ($r^2 = −0.78$) particles with depth.

↑ Variables with positive correlation with depth.

↓ Variables with negative correlation with depth.

Each PCR reaction (20 µl) included 2 µl of environmental DNA (5 ng/µl), 0.5 µl of each primer (10 µM) and 10 µl of 2× Phusion High-Fidelity MasterMix (Thermo Scientific, Waltham, MA, USA). The correct size of the amplicons was verified on an QIAxcel Advanced system (QIAGEN, Hilden, Germany), DNA and the PCR clean-up were carried out using AMPure XP beads to discard free primers and primer dimer species. In the second PCR, eight cycles attached dual indices and the Illumina sequencing adapters using the Nextera XT Index Kit. PCR barcoded amplicons were verified and purified as previously described and quantified using a Qubit 3.0 fluorometer (Life Technology, Shah Alam, Selangor, Malaysia). The individual barcoded amplicons were diluted on 10 mM Tris (pH 8.5) and pooled in equimolar concentrations (9 pM). Paired-end sequencing (2 × 300 bp) was carried out using the MiSeq platform (Illumina, San Diego, CA, USA) with a MiSeq Reagent Kit V3 (600 cycles). Sequencing was performed in the Aquatic Pathology laboratory at CINVESTAV-Mérida.

## Data analysis

Demultiplexing of the pooled amplicons and trimming the barcode and adapter sequences was performed using the MiSeq Reporter Metagenomics Workflow (*Illumina, 2014*). Reads were overlapped with the fast length adjustment of short reads to improve genome assemblies (FLASH) software (*Magoč & Salzberg, 2011*), and processed using the QIIME (version 1.9) (*Caporaso et al., 2010*) pipeline with the parameters q (phred_quality_treshold) = 20, r (max_bad_run_length,) = 3, p (min_per_read_length_fraction) = 0.75, for quality filtering. The demultiplexed sequences were clustered in operational taxonomic units (OTUs) with the 'pick_open_reference_otus.py' script at 97% of similarity using the usearch61 method

(*Edgar, 2010*) with a minimum OTU cluster size of 5. Chimeric sequences were removed with the uchime2 algorithm in the reference mode (v 9.1.13) (*Edgar, 2016*) and, the taxonomic assignment was performed by the SortMeRna (*Kopylova, Noé & Touzet, 2012*) with an *e*-value of 3 $e^{-6}$ and default parameters from QIIME, using the GreenGenes database (v13.8). An OTUs alignment was performed with the Mafft algorithm (*Katoh & Standley, 2013*) to build a phylogenetic tree using the Fasttree software (*Price, Dehal & Arkin, 2009*) for its subsequent use in the UniFrac (*Lozupone, Hamady & Knight, 2006*) distance analysis.

Community structure and composition analyses were performed by processing the OTU table in the R environment (*R Core Team, 2014*) with the Phyloseq (*McMurdie & Holmes, 2013*), vegan (*Oksanen, 2013*) and ggplot2 (*Wickham, 2010*) packages. The data set was rarefied at the depth from the smallest library (20,400). We reported the observed OTUs, the Shannon and Simpson diversity indexes ($H'$ and $D$, respectively) and the nonparametric richness from *Chao1*. The Good's coverage was calculated to corroborate the adequate sampling depth.

In order to compare the microbial community composition among shallow and deep sampling sites, a non-metric multidimensional scaling (NMDS) was plotted (*Lozupone, Hamady & Knight, 2006*; *Giloteaux, Goñi Urriza & Duran, 2010*) with the weighted UniFrac distance metric (*Lozupone, Hamady & Knight, 2006*) and a test of beta significance among groups of sample was performed using a two-sided Student's two-sample *t*-test with the 'make_distance_boxplots.py' script of QIIME, *p*-values were calculated using 1,000 Monte Carlo permutations. The physicochemical variables were fitted with the *envfit* function to the NMDS ordination to correlate them with the community composition (*p*-value <0.05 and 10,000 permutations). Physicochemical variables were also tested with a PERMANOVA analysis at a *p*-value <0.05 (Table S1). A differential abundance analysis was performed with the DESeq2 (*Love, Huber & Anders, 2014*) R library to identify the OTUs that have significant fold changes among the shallow and deep zones. The *p*-value was corrected with the false discovery rate (FDR) method (*Benjamini & Hochberg, 1995*) and a Log2 fold change plot was made with the significant OTUs at a *p*-value <0.01.

All the sequences are available at SRA site from NCBI database in the BioProject PRJNA429278 and biosample accession ID's SRR6457706 to SRR6457716 and the raw processed data in the supplementary material.

# RESULTS

## Physicochemical properties from marine sediments

The physicochemical variables and textures of the sediment samples are shown in Table 1. Redox potential ranged from −204 to 236 mV. The electronegative values were found in samples from 20 to 600 m depth, while the electropositive values corresponded to the deep sediments (2,800–3,700 m). TS was constant at depths from 20 to 600 m at a concentration of 0.05 µM, while the maximum values were detected for deep sites (0.10–0.12 µM). The percentage of TOM and the concentration of TOC varied for all sites with averages of 0.872 ± 0.22% and 0.484 ± 0.123 µM, respectively. The percentages of sand, lime and clay in the

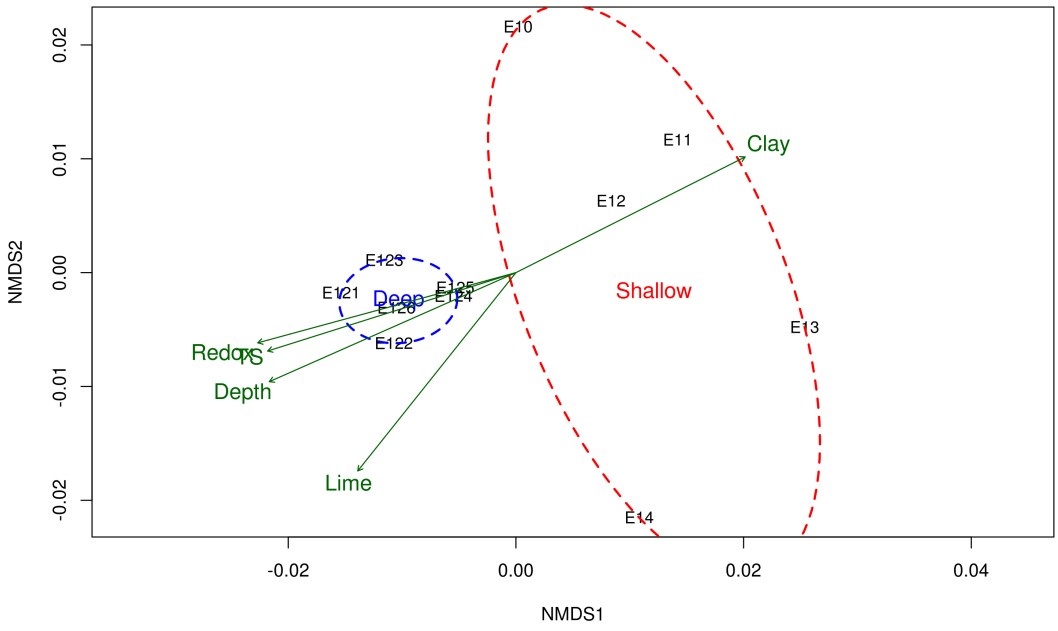

**Figure 2** **Non-metric multidimensional scaling (NMDS) of the bacterial community structure and the environmental variables.** Ordination of samples based on the weighted UniFrac distance from the community structure and their relationship with the physicochemical variables. Shallow sites included samples collected from 20 to 600 m, and deep sites from 2,800 to 3,700 m depth. Physicochemical variables related to the bacterial community structure are shown in green arrows. *TS* total sulfur concentration. NMDS stress value = 0.0684.

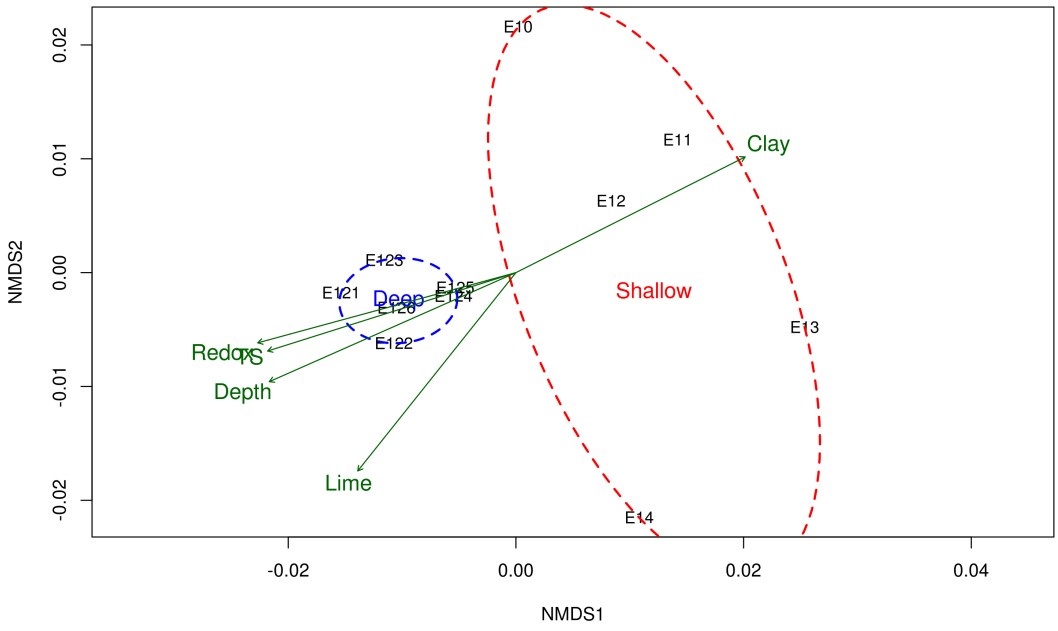

sediment samples ranged from 20.50–34.5, 51.2–70.30 and 5.9–28.4%, respectively. Values of TS, redox potential, and the percentages of lime and clay showed significant correlations with depth (*p*-value <0.01) (Table 1).

## Bacterial community structure in a depth gradient

The NMDS ordination method showed that microbial communities from shallow sites grouped separately from those from the deep sites (Fig. 2), and it was according with the test of significance were the mean of the distances of Shallow and Deep samples were different with a *p*-value <0.05. Samples taken from shallow sites displayed greater dissimilarity distance from each other, while those derived from the deep sites displayed a closer proximity to each other (Fig. 2). The environmental variables fitted on the NMDS ordination were: depth ($r^2 = 0.79$), redox potential ($r^2 = 0.77$), the concentration of total sulfur ($r^2 = 0.73$) and percentages of clay ($r^2 = 0.71$) and lime ($r^2 = 0.7$), at a *p*-value <0.05.

The observed OTUs, Shannon ($H'$) and Simpson ($D$) diversity indexes, and the nonparametric richness estimation from *Chao1*, were different among shallow and deep-sea sites (*p*-value <0.05). These ecological estimators showed higher values in the sediment samples from shallow sites (observed OTUs = 2,888.2 ± 567.88, $H' = 9.6 \pm 0.85$, $D = 0.99 \pm 0.005$ and *Chao1* = 3,791.2 ± 737.81) than those obtained from the deep ocean floor (observed OTUs = 1,884.7 ± 464.2, $H' = 7.74 \pm 1.02$, $D = 0.97 \pm 0.02$, and *Chao1* = 2,806.18 ± 589.39) (Table 2).

**Table 2  Alpha diversity estimations.** Alfa diversity data from observed OTUs, *H′* Shannon diversity index, D Simpson diversity index, and Chao1 for nonparametric richness estimation. Statistical differences were observed among sampling sites Shallow and Deep-sea (*p*-value <0.05). Percentages of Good's coverage shows the fraction of the OTUs subsampled more than once.

| Site | Sample | Depth (m) | Total counts | Observed OTUs | Ecological estimators | | | Good's coverage (%) |
|------|--------|-----------|--------------|---------------|------|------|-------|---------------------|
| | | | | | *H′* | *D* | Chao1 | |
| Shallow | E10 | 20 | 48,886 | 2,401 | 9.3 | 0.99 | 3,002.2 | 96.51 |
| | E11 | 75 | 30,290 | 3,262 | 10.1 | 1.00 | 4,141.4 | 94.83 |
| | E12 | 100 | 21,243 | 3,509 | 10.4 | 1.00 | 4,770.5 | 93.94 |
| | E13 | 200 | 35,253 | 3,081 | 9.8 | 1.00 | 3,925.3 | 95.09 |
| | E14 | 600 | 42,131 | 2,188 | 8.2 | 0.99 | 3,116.4 | 95.8 |
| Deep | E121 | 2,800 | 53,408 | 2,238 | 8.7 | 0.99 | 3,087.3 | 95.75 |
| | E125 | 3,600 | 56,735 | 1,434 | 7.0 | 0.97 | 2,059.0 | 97.36 |
| | E122 | 3,700 | 20,402 | 1,243 | 6.7 | 0.97 | 2081.6 | 97.23 |
| | E123 | 3,700 | 61,025 | 2,139 | 8.0 | 0.97 | 3,181.1 | 95.54 |
| | E124 | 3,500 | 42,768 | 1,848 | 6.9 | 0.94 | 3,009.7 | 95.80 |
| | E126 | 3,700 | 45,107 | 2,406 | 9.11 | 0.99 | 3,418.39 | 95.40 |

**Notes.**
Alfa diversity data from observed OTUs, *H′* Shannon diversity index, *D* Simpson diversity index, and Chao1for nonparametric richness estimation. Statistical differences were observed among sampling sites Shallow and Deep-sea (*p*-value <0.05). Percentages of Good's coverage shows the fraction of the OTUs subsampled more than once.

## Bacterial community composition

Microbial community analysis resulted in the detection of 25 bacterial phyla, however only 14 were ≥1% in relative abundance (Table S2). *Proteobacteria*, represented by *Gamma-*, *Alpha-* and *Deltaproteobacteria* classes, were dominant in all the analyzed samples (Fig. 3). Nevertheless, *Bacteroidetes*, *Acidobacteria*, *Chloroflexi*, *Nitrospirae*, *Planctomycetes*, *Gemmatimonadetes* and the *NC10* phyla were also well represented in the samples. Two archaeal phyla, corresponding to the *Euryarchaeota* (*Thermoplasmata* class) and *CreNarchaeota* including *Miscellaneous CreNarchaeota Group* (MCG) and *ThauMarchaeota* classes (based on the GreenGene database), were also detected in low abundances (0.3 to 1.2%) (Fig. 3). At lower taxonomic levels, 51 families were detected, from which *Piscirickettsiaceae*, *Rhodobacteraceae*, *Flavobacteriaceae*, *Syntrophobacteraceae*, *Thermovibrionaceae*, *Desulfobacteraceae*, *Colwelliaceae*, *Marinicellaceae*, *Alcanivoracaceae*, *Colwelliaceae*, and *CeNarchaeaceae* were among the families with higher relative abundances in the samples (Fig. S1A). Moreover, 23 genera were identified at >1% of relative abundances (Fig. S4). *Desulfococcus*, *Alcanivorax*, *Fulvivirga*, *Amphritea*, BD2-6, *Shewanella*, ZD0117 and *Nitrospina* were the better represented in the samples (Fig. S4).

Differential abundance analysis identified 147 OTUs with significant fold changes between the deep and shallow zones, 107 from shallow and 40 from deep sites (Fig. 4A). From the shallow samples, most of these OTUs belonged to *Delta-*, *Alpha-* and *Gammaproteobacteria* (38, 24 and 18, respectively), while the rest of the OTUs were distributed in 13 classes of 10 phyla (Fig. 4A). In deep samples, all of the 40 OTUs belonged to *Gamma-*, *Alphaproteobacteria*, *Bacteroidetes* and *Nitrospirae* (27, six, six and one, respectively) (Fig. 4A). The relative abundances of the 147 OTUs with significant fold changes constituted approximately 10 to 40% of the total microbial communities (Fig. 4B).
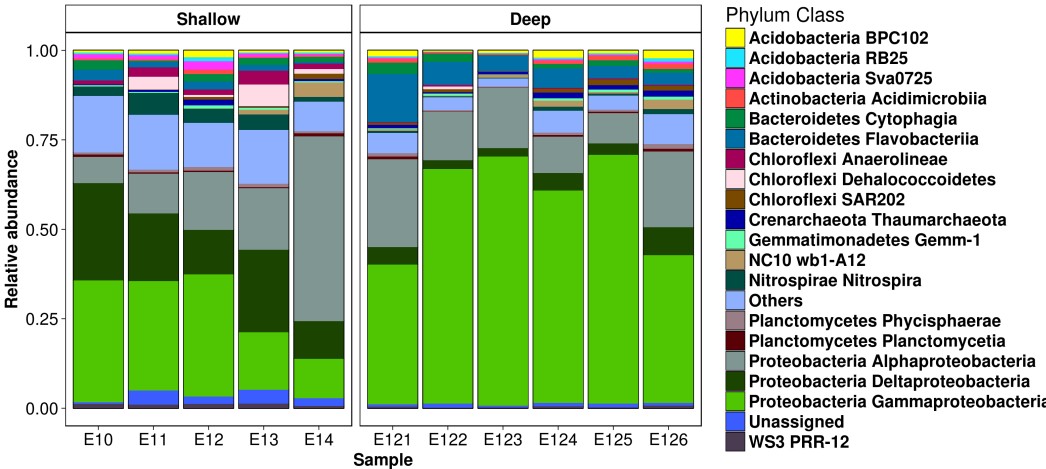

**Figure 3  Relative abundances from bacteria inhabiting sediments in the Mexican Region in the northwestern GM.** Taxonomic diversity at phylum and class levels from shallow and deep samples. Shallow sediment samples retrieved from 20 to 600 m depth, and **deep** sediment samples retrieved from 2,800 to 3,700 m depth. Only the 20 most abundant classes are shown. The remaining classes were agglomerated in the "Others" category.

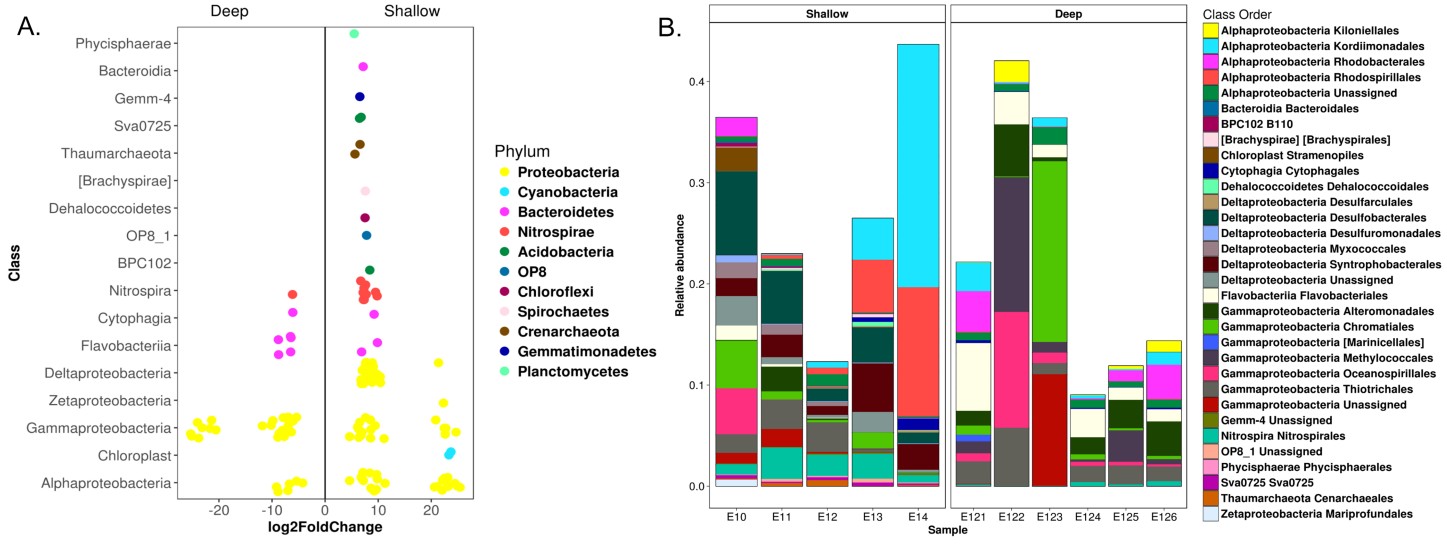

**Figure 4  Differential OTUs abundance among shallow and deep communities from sediment samples in the Mexican northwestern Gulf of Mexico.** (A) OTUs with significant fold changes among the community composition from shallow and deep zones. Circles in colors defines the phyla of the OTUs. (B) Relative abundances of the OTUs with significant fold changes at a class-order taxonomic level.

*Kordiimonadales, Rhodospirillales, Desulfobacterales, Syntrophobacterales* and *Nitrospirales*, were the orders with highest relative abundances for shallow samples whilst the orders *Chromatiales, Oceanospirillales, Methylococcales, Flavobacteriales* and *Rhodobacterales* were the best represented orders for deep samples (Fig. 4B). At the genus taxonomic level,

differential abundance analysis allowed to identify *Marinicella*, *Alcanivorax*, *Shewanella* and *ZD0117* with significant fold changes for deep samples; and *Nitrosopumilus*, *GOUTA19*, *LCP-6*, *BD2-6*, *Amphritea*, *Desulfococcus* and *Mariprofundus* for shallow sites (Fig. S5).

## DISCUSSION

### Physicochemical properties of sediment samples

Physicochemical properties (e.g., temperature, pH, redox) in marine sediments commonly change according to water depth (*Miller & Wheeler, 2012*). In this study, lime percentage, redox potential and total sulfur were positively correlated to depth, while the clay particles percentages were negatively correlated with depth (Table 1). The electronegative redox values measured in the first 5 cm of the sediments in all shallow sites (Table 1), suggested that oxygen was depleted by microbial respiration as common reported for this layer in shallow marine sediments (*Aranda et al., 2015*). Electronegative redox values in the shallow samples were likely enhanced at higher fine-grained clay content, since this diminishes the sediment permeability as reported elsewhere for sediments in the continental platform (*Probandt et al., 2017*). In contrast, the electropositive redox values detected in the first 5 cm of the sediments in all the deep sites (Table 1) suggested oxidizing conditions. This condition has also been observed at different latitudes in the GoM (*Deming & Carpenter, 2008*) due to both the circulation of deep water masses carrying oxygen (*Jochens & DiMarco, 2008*), and low rates of oxygen consumption (*Rowe et al., 2008*).

Total organic matter percentage (TOM%) was determined to be below 2% for all the sediments analyzed (Table 1). These low organic contents are similar to those reported in studies of two onshore-offshore transects in the Northwestern GoM (*Goñi, Ruttenberg & Eglinton, 1998*). Shallow sites likely receive inputs from terrestrial organic matter (*Balsam & Beeson, 2003*), and yet these sediments presented similar values of TOM than those detected for deep sites (Table 1). Previous studies have reported high production and consumption rates of oxygen measured for shallow sediments in Northeastern GoM (*Mills et al., 2008*). TOM values determined in this study for shallow sites are in concordance with previous reports for the GoM (*Balsam & Beeson, 2003*). We hypothesize that these values are related to the consequence of high rates of microbial metabolism in those same zones.

### Microbial ecology assessment

The effect of storage temperature on microbial community structure has been previously studied, and it is well recognized that microbial metabolism could keep on going at $-20\,°C$; however, the survival at these subzero temperatures requires several genetic and physiological strategies that exclusively cold-adapted microorganisms (permafrost and seasonally frozen soil microbial communities) have developed (*Jansson & Tas, 2014*; *Boetius et al., 2015*). In the present study, microbial communities from marine sediments living at temperatures ranging from 4 to 27 °C were analyzed. Thus, we considered that changes in the microbial community composition due to the storage temperature ($-20\,°C$) were unlikely, because the sediments were immediately frozen at $-20\,°C$ on board and they were never thawed during the transport neither in the laboratory before DNA extraction, as

has been recommended in previous studies (*Tedjo et al., 2015*). The storage of the samples at −20 °C is a standard procedure commonly used for microbial community analyses from marine sediment (e.g., *Bienhold et al., 2016*; *Probandt et al., 2017*).

Diversity indices are commonly used in microbial ecology studies to understand the links between the environmental conditions and the community (*Kimes et al., 2013*; *Bargiela et al., 2015*). It has been suggested that microbial richness and diversity could be an expression of environmental variation correlated to energy sources and temperature (*Santelli et al., 2008*; *Rosano-Hernández, Ramírez-Saad & Fernández-Linares, 2012*; *Bargiela et al., 2015*). Our results revealed higher richness and diversity in shallow sites compared to deep sites (Table 2). This can be a result of more dynamic shallow environments, characterized by strong physical mixing and seasonal variation. In shallow environments the interaction among atmosphere, land and ocean increase environmental variation, and therefore higher microbial diversity, as reported for coastal environments elsewhere (*Gobet et al., 2012*). In contrast, lower richness and diversity in the deep sites could be the result of relatively constant environmental conditions. Microbes from the abyssal plains environment are able to overcome the extreme conditions of temperature, pressure, oligotrophy and darkness (*Jochens & DiMarco, 2008*) but are subjected to far less environmental variability.

It is well recognized that marine sediments are dynamic environments shaped by interactions among biotic and abiotic processes, including the redox reactions (*Wasmund, Mußmann & Loy, 2017*). In the present study, the electronegative redox values detected in the first 5 cm of the sediments for shallow sites (Table 1), suggested high microbial respiration or anoxic/reducing conditions (*Valdes & Real, 2004*; *Aranda et al., 2015*). However, genera known to be both aerobic (e.g., *Maricinella and Nitrosopumilus*) and anaerobic (e.g., *Amphritea, Desulfococcus, GOUTA-19, LCP-6,* and *Pseudidiomarina*) were found in shallow samples (Fig. S4). This suggested the co-occurrence of different metabolisms in these sites, in accordance with previous reports for other shallow marine sediments (*Jørgensen, 1977*; *Mills et al., 2008*; *Acosta-González, Rosselló-Móra & Marqués, 2013*; *Tolar, King & Hollibaugh, 2013*; *Cerqueira et al., 2015*).

Electropositive redox values detected in the deep sites (Table 1), suggested oxidizing conditions at those water depths, likely because the upper sediment layers remain oxygenated due to relatively high oxygen concentrations in deep waters of the GoM (*Jochens & DiMarco, 2008*) as well as low rates of sediment community oxygen consumption previously reported in the deep GoM (*Rowe et al., 2008*). In these oxidizing deep sediments, genera known to be aerobic and facultative anaerobic bacteria, such as *Shewanella, Alcanivorax, Maricinella, Nitrospina* and *ZD0117* (*Alteromonadacea* e) were well represented (Fig. S4). Thus, these results suggest that aerobic (and facultatively anaerobic) lifestyles seem to be favored in abyssal sediments of the studied region.

In this study, water depth was highly correlated with microbial community structure (Fig. 2). It is known that water depth and its dependent variables, such as temperature and pressure, are among the most important factors explaining variation in the bacterial community composition of seafloor sediments (*Orcutt et al., 2010*; *Bienhold et al., 2016*). As an example of this, our results showed a relatively high abundance of psychropiezophilic

organisms of the genera *Colwellia* and *Shewanella* for the deep sites (Fig. S2 and Fig. S4), where conditions are favorable for the development of these microbial metabolisms (*Nogi et al., 2004*).

## Detection of phylotypes putatively related to hydrocarbons degradation

Due to the presence of natural hydrocarbon effluents and the oil spills which have occurred in the GoM (e.g., DWH spill), the potential effects of hydrocarbons on microbial community structure has been extensively studied there (*Rosano-Hernández, Ramírez-Saad & Fernández-Linares, 2012*; *Orcutt et al., 2010*; *Kleindienst et al., 2015*). Recent investigations have provided insights about microbial populations responding to the presence of hydrocarbons, highlighting microorganisms capable of using these organic compounds as carbon source (*Kostka et al., 2011*; *Redmond & Valentine, 2011*; *Mason et al., 2012*). In this work, we report the occurrence of several genera, such as *Shewanella*, *Alcanivorax*, *Pseudoalteromonas* and *Phaeobacter* (Fig. S4), which have been previously associated with hydrocarbon consumption (*Beazley et al., 2012*; *Mason et al., 2014*; *Barbato et al., 2016*; *Liu, Bacosa & Liu, 2017*).

In addition, this work reports for the first time *NC10* and *Kordiimonadales* phylotypes for sediments of the GoM (Fig. 3 and Fig. S1), that based on the available information, are capable of anaerobic methane oxidation with nitrite denitrification, and aerobic degradation of alkanes and polycyclic aromatic hydrocarbons, respectively (*Kwon et al., 2005*; *Xu et al., 2011*; *Math et al., 2012*; *Zhou & Xing, 2015*; *Padilla et al., 2016*). The *Kordiimonadales* phylotype was even detected by DGGE band sequencing approach (Fig. S2B), supporting their relatively high abundances in the analyzed samples.

The potential capability to degrade aromatic compounds, such as nitrotoluene, ethylbenzene, chlorocyclohexane and fluorobenzoate were predicted based on trait modeling software (PICRUSt) (Fig. S3). This also suggests that microbial metabolisms related to the hydrocarbon degradation might be well distributed in the Mexican region of the PFB and could contribute with the bioremediation in case of hydrocarbon contamination due to forthcoming oil exploration and extraction activities in this area.

## CONCLUSIONS

To our knowledge, this is the first study to report differences in abundances and composition of the microbial communities inhabiting surficial sediments in a water depth gradient in an important province for oil extraction in the Mexican region of the Gulf of Mexico. In this study, the combination of nucleic acid–based molecular methods and physicochemical measurements allowed the detection of changes in the structures of microbial communities which were mainly related to the redox potential, total sulfur concentration, grain size, as well as depth and the variables that change with it, such as temperature and pressure. Our findings indicate that shallow microbial communities are taxonomically richer than communities in the deep sediments. Bacterial phylotypes putatively related to hydrocarbon degradation appear to be well represented for all the analyzed samples and could ameliorate future anthropogenic oil spills in this region of

the GoM. These results contribute to the current knowledge of the environmental and biological dataset of the Perdido Fold Belt, which will be useful for further monitoring of this area in the Gulf of Mexico.

## ACKNOWLEDGEMENTS

We are thankful to David Valdes and Silvia Guadalupe Granados for providing the physicochemical data; and to Abril Gamboa-Muñoz for the preparation of Fig. 1. Gregory Arjona and Francisco Puc were responsible of the sampling during the oceanographic campaign. We are very appreciative of many helpful discussions with Santiago Cadena and Alejandra Escobar-Zepeda. The computational resources used for bioinformatics analyses at CINVESTAV-Mérida were provided by Emanuel Hernández-Núñez.

### Funding

This research was provided by ''Biotechnology of marine organisms'' awarded by the National Council of Science and Technology of Mexico (CONACYT) project 15689 - 2014 and by the Marine Resources Department at CINVESTAV Merida. CONACYT awarded Ma. Fernanda Sánchez-Soto Jiménez with Master and PhD Scholarships. The necessary infrastructure to complete this investigation came from CONACYT –Mexican Ministry of Energy –Hydrocarbon Trust, project 201441; and from CONACYT 251622 - 2015 received by José Q. García-Maldonado. The funders had no role in study design, data collection and analysis, decision to publish, or preparation of the manuscript.

### Grant Disclosures

The following grant information was disclosed by the authors:
National Council of Science and Technology of Mexico (CONACYT): 15689-2014.
Marine Resources Department at CINVESTAV Merida.
CONACYT –Mexican Ministry of Energy –Hydrocarbon Trust: 201441.
CONACYT: 251622 - 2015.

### Competing Interests

Samples were collected during ''Campaña Oceanográfica y Manifestación Ambiental Modalidad Regional del Golfo de México Aguas Someras, Profundas y Ultra Profundas Región Norte, 2014'' founded by PEMEX-PEP.

### Author Contributions

- Ma. Fernanda Sánchez-Soto Jiménez conceived and designed the experiments, performed the experiments, analyzed the data, prepared figures and/or tables, authored or reviewed drafts of the paper, approved the final draft.
- Daniel Cerqueda-García and Jorge L. Montero-Muñoz analyzed the data, prepared figures and/or tables, authored or reviewed drafts of the paper, approved the final draft.

- Ma. Leopoldina Aguirre-Macedo conceived and designed the experiments, analyzed the data, contributed reagents/materials/analysis tools, authored or reviewed drafts of the paper, approved the final draft.
- José Q. García-Maldonado conceived and designed the experiments, performed the experiments, analyzed the data, contributed reagents/materials/analysis tools, prepared figures and/or tables, authored or reviewed drafts of the paper, approved the final draft.

## DNA Deposition

The following information was supplied regarding the deposition of DNA sequences:

All the sequences are available at SRA site from NCBI database in the BioProject PRJNA429278 and biosample accession ID's SRR6457706 to SRR6457716.

## Data Availability

The raw data are provided in the Supplemental File.

## Supplemental Information

Supplemental information for this article can be found online at http://dx.doi.org/10.7717/peerj.5583#supplemental-information.

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
