# Peer review of "Assessment of the bacterial community structure in shallow and deep sediments of the Perdido Fold Belt region in the Gulf of Mexico"

_PeerJ, doi:10.7717/peerj.5583_

## Round 0.1 · original submission · Major Revisions

This study study samples collected along a transect from shelf to deep waters int he Gulf of Mexico. They analyzed 11 samples for basic chemistry and 16S rRNA diversity. this is a small dataset, but baseline datasets are valuable and data from this region of the Gulf is rare. However, both reviewers agree that even if valuable, the manuscript needs an english review and have problems drawing the conclusions to far. One of the reviewers points out that the authors make big assumptions about the predicted metabolic profiles and they are not taking into account the pitfalls of their chosen method (PICRUST), they are placing this as the main result but is a weak one, with any experimental support validating their assumption. I suggest to tone down the conclusions related to the metabolic predictions, which are just an educated guess.

Reviewer 1 ·

Basic reporting

The manuscript presented by Sánchez-Soto and collaborators is a 16S rRNA massive sequencing survey for a depth and longitudinal transect (20-3700 m) in waters in the northwestern Gulf of Mexico within the Mexican region. The supplied background is enough although it would be enriched if detailing findings from other similar works like the related from the deep-horizon spill.

The manuscript presentation is overall good. However, it needs some extra work for some vague phrases all around the text that could be fulfilled with references and declaring missing values.

Authors make big assumptions about the predicted metabolic profiles and they are not taking into account the pitfalls of their chosen method (PICRUST), they are placing this as the main result but is a weak one, with any experimental support validating their assumption. I suggest to tone down the conclusions related to the metabolic predictions, which are just an educated guess.

Experimental design

The authors are short with methodological descriptions, which make the work impossible to be replicated by independent groups.

Experimentally, I have important concerns about your sampling methods where you state that you freeze at -20ºC the samples (L78). How much time left between the sampling, freezing, and DNA extraction? This storage temperature could modify your community profiles as microbial metabolism could keep on going at this temperature (Price & Sowers 2004; Panikov et al., 2006).

Is there any chance that you have water/sediment temperature to be added to your Table 1?

DNA extraction: How much sediment are you using for each extraction?

Did you perform MiSeq sequencing on your own or are you paying for the service? Please declare who did the sequencing.

For the data analysis methodology, there are several missing descriptions, and I recommend to the authors to prepare supplementary information material regarding your bioinformatic and statistical procedures. You should consider using jupyter notebooks, or any markdown log available and upload your parameters and scripts to GitHub, figshare or any other public repository.


Price, P. B., & Sowers, T. (2004). Temperature dependence of metabolic rates for microbial growth, maintenance, and survival. Proceedings of the National Academy of Sciences of the United States of America, 101(13), 4631–6. http://doi.org/10.1073/pnas.0400522101
Panikov, N. S., Flanagan, P. W., Oechel, W. C., Mastepanov, M. A., & Christensen, T. R. (2006). Microbial activity in soils frozen to below −39 °C. Soil Biology and Biochemistry, 38(4), 785–794. http://doi.org/10.1016/J.SOILBIO.2005.07.004

Validity of the findings

Findings seem legit. Metabolic predictions should be toned down and identify it as speculative in the discussion, not as hard result.

Could not assess the validity of the findings because of the lack of the raw data, specifically OTU tables with their assigned taxonomy, metadata files, and the metabolic predictions. This should be added as supplemental information for the manuscript and publicly available. This could be overridden with a detailed methodologic description as suggested before.

Additional comments

L29 What is statistically more abundant?
L41 Wrong citation? What in the world is chemical homeostasis? Leloup et al. 2009 do not even say homeostasis in his manuscript!
L45 microbial (?) community
L59 Missing reference (oil spill 2010).
L70-71 You are just predicting, guessing to close related sequenced strains, what do you know about pan-genomic variability?
L102 16F and 16R? please write 5'-ATCG-3' pair of primers used (complete).
L114-115 Where did you sequence your samples?
L121 What are q, r and p parameters?
L126 What parameters did you use to compare against Greengenes DB?
L132 Normalized? Do you mean rarified? Subsampled?
L145-154 NOT reproducible without further details about the parameters used. How many enzymes are involved in hydrocarbon and xenobiotic degradation within PICRUST/KEGG? Did you just perform this analysis?
L169 What is clearly distinguished?
L173 NMDS what is the stress value? Is it a robust ordination?
L261 incomplete comparison, sentence (differentially abundant)
L263-L265 Incomplete idea, check sentence structure. Missing references?
L268-272 I don't get your idea is quite confusing
L309 References missing
L309-312 I don't understand your sentence. Rhizobiales (in your samples?) what is a complex diversity?
L319-323 Did you found any correlation between your physicochemical variables with shallow samples? Have you tired Mantel tests?
BASELINE. You are giving one snapshot, I think it is pretty presumptuous to establish a baseline with 11 samples in just one season. Tone down baseline term use of the manuscript.

Reviewer 2 ·

Basic reporting

1. the English is not bad (certainly better than my Spanish), but could use help from an editor at PeerJ or a professional English language editor. there are places thorughout with usage patterns that are just a little different than what i would expect in a scientific publication, and this is beyond the duties a reviewer should be addressing

2. there are issues throughout with references cited that do not pertain to the preceding statement. i list specific examples in the General Comments below, but this was very frustrating, especially since i know many of the papers cited and was confused to see them cited where they were. this will need to be fixed, all references should be checked and verified by the authors to be necessary and accurate.

3. some of the figures can be improved, comments are below. this is not a fatal flaw, but i feel that improvement will really enhance the manuscript.

Experimental design

generally acceptable, however please add references for the TOC and TOM methods described, there is a brief summary of the method, but no citation for what is a standard method that should be cited.

Validity of the findings

this is the strongest area of the manuscript - this is a baseline study for an underexplored region of the Gulf of Mexico, and i believe that has merit. the sampling strategy is well designed, and the authors treat the data well.

Additional comments

Summary:
the authors here present microbial diversity and correlations to some environmental variable for surface sediments collected along a transect from shelf to deep waters int he Gulf of Mexico (for which the acronym used in the manuscript should be GoM or GOM, not GM). they analyzed 11 samples for basic chemistry and 16S rRNA diversity. this is a small dataset, but baseline datasets are valuable and data from this region of the Gulf is rare. i appreciate that the authors generally do not try to overstate what data they have or what it can say, but i think there could be more work put into the deep versus shallow comparison, and especially comparison with any previous data. in addition, i encourage them to look deeper into PICRUSt to determine if it can really be used with this dataset. at this stage, i believe the manuscript requires major revision before it can be considered for publication, but i also believe that the authors have a publishable dataset that can be appropriate for publication with the appropriate revisions.

Specific comments:

Abstract:
please remove the sentence: "These microbial communities were composed of more than 20 phyla, being Proteobacteria the dominating phylotypes in all the analyzed samples." this is overly broad and unsurprising enough to not warrant mention in the abstract. in general, the language in the abstract is poor and needs to be revised. the remainder of the manuscript is better, but also needs review from an English language expert/native speaker.

Introduction:
Line 45: the Santelli reference is not appropriate here, this is a study of microbiology on seafloor exposed basalts, it doesn't get much into environmental factors regulating community structure

Line 49: here and forward (i will not continue to edit), you must use "Alpha-". Also, it should be Gammaproteobacteria (or whichever class you are identifying), not Gamma-proteobacteria

Line 56: there are NO hydrothermal vents in the Gulf of Mexico - the vents discussed in this reference are from the Galapagos spreading center. remove this citation.

Line 58: there is SO MUCH work related to the DwH spill, you should cite some of these papers. anything from Mandy Joye and Joel Kostka's groups will suffice, i know you cite a review from them later but they have much more and some should be included here.

Line 71:
change "explored" to "predicted based on trait modeling software"


Methods:
Line 82: must add citations for methods used in TOM and TOC analysis


Results:
Line 182: first of all, you are referencing phyla for a figure that shows class, although the key gives phyla and class, so maybe this is ok. second, there are too many classes displayed and several have the same colors - change this figure to display only classes >1% relative abundance or some other cutoff that allows you to discuss the groups you want but reduces the number in the plot. you can (and should) include a supplementary table of the OTU distribution per sample with OTU taxonomy indicated and readers that want finer details can go there to learn about their favorite minor community members, some of which is already in your first supplementary figure

Line 204: which hydrocarbonoclastic communities? this comes out of nowhere, with no details on how you determined which communities are hydrocarbonoclastic, and which members of the communities putatively are associated with this metabolism. you need a few sentences or a whole paragraph explaining why you care about hydrocarbonoclastic microbes in this region, and then identifying which of your iTags may be indicative of that metabolism. then you can back it up with the PICRUSt analysis (assuming low NSTI values)

Line 204: PICRUSt works best for human microbiome studies - report your weighted NSTI values, and if they are close to or above 0.20, then these calculations are not meaningful. also, why did you pick those putative metabolisms? was this a fishing trip or do you have hypotheses you are trying to test?


Discussion:
Line 244: diversity indices are HIGHLY influenced by sampling depth - you can only make this comparison if you have reanalyzed the data from the other papers, and i do not believe you did that here

Line 246: th Kimes citation is incorrectly formatted in the reference list

Line 261 - this sentence should read "The detection of high abundance of lineages known to be anaerobic..."

Line 277 - change to "The phylotypes detected chich have been previously described as aerobic include Flavobacteriaceae..."

Line 290: no idea why you are citing Redmond and Valentine here, this is not appropriate for what you are looking for (your deep sites much deeper than the DwH hydrocarbon plume they were studying)

Line 292: the study of hydrocarbonoclastic microorganisms has gained in relevance, the organisms themselves are not aware of their own relevance. this is nitpicky, but will improve the language

Line 298: the Math 2012 and Fuhrman 2015 references are incorrect for this statement and need to be removed here. throughout the paper, your references are frustratingly inaccurate, a reference must refer to the statement it is cited after, Fuhrman 2015 is a review paper about microbial ecology and network methods, which has nothing to do with GoM hydrocarbons. and the Math 2012 paper describes a bug from KOREA, yet you reference it to illustrate GoM work. frustrating. at least the Math 2012 paper belongs cited the next time it appears

Line 309: about aromatic compound degradation - need a citation for this. again, you are not pushing your PiCRUST interpretation too far, but be sure the NSTI scores are appropriate enough to use with your samples. i have had to abandon a PiCRUST analysis in one of my own papers because the scores were too high and therefore the bugs in the PiCRUST database do not well represent those in my samples (possibly true for you as well)

Figures 3 and 4 - the color schemes are very difficult to interpret, i think these figures should be redesigned to display ONLY what you want the reader to know. with so many colors, and the keys so (so) small, i cannot figure out which color bars refer to which groups

Figure 4, only - i'm not sure panel A is informative since there's no indication of the OTU taxonomy beyond class. maybe you can remove this one, or move to the supplement. i actually think this aspect (differential abundance for deep vs shallow samples) is probably most interesting and would like to see it discussed a bit more in the text.

while the transition in microbial diversity with depth from shelf to deep sea may not be well documented in this area, i bet there have been other studies looking at that question from other regions. i can't name them off the top of my head, but encourage the authors to seek them out and cite them in their intro, and also make comparisons of their data to any previous data in the discussion. additionally, there is a well established and reviewed literature on this transition in macrofauna - see chapter 13 in Miller and Wheeler 2012, Biological Oceanography (textbook), for a discussion and many references. please cite some of these classic papers, it gives context to your own work.

---

## Round 0.2 · Minor Revisions

Please address the remaining minor comments from the reviewers

Reviewer 1 ·

Basic reporting

no comment

Experimental design

no comment

Validity of the findings

no comment

Additional comments

The manuscript has substantial improvements from the previous versions. However, there are some considerations still to be fulfilled before acceptance. First is that raw data allows more reproducibility than the previous review, although the analytical and statistic methodology is still vague. The recommendation is to take a look for the following URL for guidance in the elaboration of a supplementary methods file including the complete methods files:

http://nbviewer.jupyter.org/github/biocore/qiime/blob/1.9.1/examples/ipynb/illumina_overview_tutorial.ipynb

Secondly, some of the observations of the previously reviewed manuscript are discussed in the response letter but not incorporated into the current version manuscript. Specifically:

"How much time left between the sampling, freezing, and DNA extraction? This storage temperature could modify microbial community profiles as microbial metabolism could keep on going at this temperature (Price & Sowers 2004; Panikov et al., 2006). "

Price, P. B., & Sowers, T. (2004). Temperature dependence of metabolic rates for microbial growth, maintenance, and survival. Proceedings of the National Academy of Sciences of the United States of America, 101(13), 4631–6. http://doi.org/10.1073/pnas.0400522101
Panikov, N. S., Flanagan, P. W., Oechel, W. C., Mastepanov, M. A., & Christensen, T. R. (2006). Microbial activity in soils frozen to below −39 °C. Soil Biology and Biochemistry, 38(4), 785–794. http://doi.org/10.1016/J.SOILBIO.2005.07.004

In the current authors' answer, It is like they just nod and then ignore the observations, stating that "microbial communities have evolved to develop the mechanism of cold adaptation" without any supporting reference, It is not even stated that the samples remained for five months at -20ºC.

Please include this into the methods and discuss it, why the authors did not snap-freeze the samples, which is the standard procedure, please take note of the following references as well:

http://journals.plos.org/plosone/article?id=10.1371/journal.pone.0126685
https://www.nature.com/articles/nrmicro3262

Authors answer only states that other studies have done it that way.

Qiime is missing its reference; you are just mentioning the URL.

Reviewer 3 ·

Basic reporting

The language used in the manuscript is passable. There are word choices that are not typical and make some sentences more challenging to read however, this does not significantly affect reader understanding. I was unable to open the OTU table in the supplemental files. Perhaps this was an issue on my end but, I usually have no trouble reading .biom files. Due to this issue, I was not able to access the validity of some of the data presented.

Experimental design

The experimental design and analysis methods were sufficient for the author's objectives.

Validity of the findings

I feel that the taxonomic analysis of the data is a little superficial. The authors limit themselves to describing the microbial community structure at very broad taxonomic levels.
This causes the authors to make very general and highly speculative statements throughout the discussion which are poorly supported. I would recommend that the authors take the time to do a detailed assessment of the community at the genus level and highlight those OTUs that appear to be of particular interest. If necessary, the authors can cross check the taxonomic classification of their OTUs against multiple databases. This will improve the overall quality of the manuscript and lend more support for some of the claims the authors make related to potential metabolic characteristics.

Additional comments

The manuscript presents a description of the sediment microbial community structure along a transect with varying water column depths. Although the data set is relatively small (11 samples) the is sufficient to begin to address the research question. Overall, the manuscript is suitable for publication, but I think the authors can do more with the data on hand to improve the quality of the manuscript.

Abstact
L28: change “depth gradient” to “water column depth gradient”.
Introduction
L 59-64: These statements are not very informative.
L77: change notoriously to notably
Material and Methods:
L149: How long were paired end reads (e.g. 300 bp?)
Data analysis: please provide details about number of reads from the sequencing effort, number of reads that passed QC, etc. This can be provided in a small supplemental table.
L163: change assignation to assignment
L178: specify package that includes envfit function
Results
L204-205: Are the shallow and deep communities statistically different from one another?
L224-224: provide range of relative abundance for these archaeal groups since they do not show up on your figures.
L230-241: It would be nice to know what these OTUs actually were at a genus level. You description and figures lump everything together at a high taxonomic level which make if challenging to draw any significant meaning from this analysis.
Discussion
L246-247: sentence confusion. I am assuming you are talking about temperature, oxygen concentration, etc as the variables that correlate with water depth, but these are physicochemical properties. Recommend rewording.
L249-251: It is important to note that it looks like your redox values were determined based on a bulk sample of ~4cm. It is highly likely that the whole sample was not anoxic, but rather that oxygen was quickly depleted after the first few mm to cm as is common in marine sediments. Unless you have evidence that the bottom waters were hypoxic, this statement needs to be refined. Also, do your redox profiles show any correlation with TOM or TOC?
L255: define deep sediments. Are you referring to the water column depth or the sediment depth profile.
L262-264: is your site in an area where water currents and river runoff make it likely to be impacted by terrestrial input ?
L266: the TOC values that you report are not low but very standard for the GoM (e.g. Escobar-Briones and Garcia-Villalobs 2009).
287-294: see comments relating to lines L249-251. This is not surprising giving the lack of depth resolution in your redox measurement. Also, several of the groups that you pointed out as being aerobic do have anaerobic members. Please be cautious about drawing conclusions about metabolic characteristics from such broad taxonomic groupings.
L299: Here you suspect low rates of sediment oxygen consumption but on line 267 you suggest high rates of microbial metabolism. General high microbial metabolism equates to high oxygen consumption rates…
L300-L303: see comments from L287-294.
L308-310: There are LOTS of member of the Altermonadales that are neither psychrophilic nor piezophilic. Do you have evidence that individual OTUs within this family are related to known psychropiezophiles? Again please refrain from making conclusions about physiology and metabolism based on such broad taxonomic groups.
L328: remove strictly. This suggests all member of the Kordiimonadales have these metabolisms which is not demonstrated.

---

## Round 0.3 · accepted · Accept

The authors have made all the minor revisions asked by the reviewers and I think it is ready to be published, it is a good study.

#